# Psychometric Validation of a Standardized Instrument for Assessing Food and Nutrition Security Among College Students

**DOI:** 10.3390/nu17152514

**Published:** 2025-07-31

**Authors:** Rita Fiagbor, Onikia Brown

**Affiliations:** Department of Nutritional Science, Auburn University, 260 Lem Morrison, Auburn, AL 36849, USA

**Keywords:** nutrition security, food security, reliability, validation, survey

## Abstract

**Background/Objective**: Food insecurity refers to social or economic challenges that limit or create uncertainty around access to enough food. Among college students, food security status is usually determined with the USDA 10-item Food Security Survey Module, which has not been validated for this population. Nutrition security refers to consistent access to food and beverages that promote well-being, prevent disease, and emphasize equitable access to healthy, safe, and affordable foods. Currently, there is no standardized measure that assesses food and nutrition security tailored to the unique experiences of college students. This study aims to evaluate the validity and reliability of a newly developed College Student Food and Nutrition Security Survey Module (CS-FNSSM). **Methods**: A mixed-methods approach that combined an online survey with semi-structured cognitive interviews. Participants were students aged 18 and older from U.S. public universities. Quantitative data were analyzed using RStudio (version 4.4.1), and interview transcripts were thematically analyzed. **Results**: Survey responses were collected from 953 participants, including a subset of 69 participants for reliability testing and 30 participants for cognitive interviews. Rasch analysis showed good item performance and structural validity. The CS-FNSSM demonstrated strong sensitivity (89.09%), specificity (76.2%), moderate test–retest reliability (0.59), and good internal consistency (Cronbach’s alpha = 0.79). Qualitative findings confirmed participant understanding of the items. **Conclusions**: The CS-FNSSM effectively identifies food and nutrition insecurity, with nutrition security emerging as a key issue. Addressing both is crucial for promoting the overall health and well-being of college students.

## 1. Introduction

Food insecurity, defined as the lack of affordable and consistent access to adequate food, has been linked to adverse physical, mental, and academic outcomes [1,2,3,4,5,6]. In contrast, nutrition security refers to consistent access to food and beverages that promote well-being, prevent disease, and emphasize equitable access to healthy, safe, and affordable foods for all [2,7]. College students often face unique challenges in accessing sufficient and nutritious food; thus, food and nutrition security is crucial for their health and well-being. Food insecurity prevalence among college students ranges from 14% to 50% [8,9,10,11,12,13,14]. The increasing cost of higher education, coupled with rising living expenses, has exacerbated food insecurity among students, potentially impacting academic performance, mental health, and overall well-being [15].

Given the significant implications of food insecurity for student success, accurate evaluation of food security estimates is essential for informing policy and intervention efforts. Despite the growing recognition of food insecurity among college students, currently, there is no validated instrument specifically tailored to this population. The U.S. Department of Agriculture’s Food Security Survey Module (USDA-FSSM) is widely used to assess food security in the general population [16]. However, its applicability and validity to college students have been questioned [17,18,19]. College students face unique challenges with food insecurity compared to other groups. The transitional nature of college life marks a critical period between adolescence and adulthood, during which students begin to experience greater independence while still relying on support from family or institutions in certain areas. This formative period often brings uncertainty surrounding one’s identity, future goals, and personal values, as well as rapid changes in living arrangements, social networks, and responsibilities. As students adjust to these shifts, they may become more vulnerable to food insecurity, mental health challenges, and risky behaviors [20]. Factors like campus meal plans, fluctuations in financial aid, and restricted access to cooking facilities contribute to their distinct experiences [17,21,22].

Previous research has demonstrated that the standard food security measures (USDA-FSSM) may not fully capture college students’ food security experiences. For example, Laska et al. [23] emphasized the need for a nationally representative estimate and a valid measurement of food insecurity that applies well to college populations, as traditional survey items do not account for the buffering effects of university food assistance programs or intermittent parental support. Additionally, standard tools may not adequately capture coping mechanisms unique to college students, such as skipping meals to afford textbooks or relying on on-campus food pantries [24]. To address the aforementioned concerns, this study aims to validate a food and nutrition survey tailored for college students using the Rasch model through scaling analyses and cognitive interviews. Specifically, this study aims to (1) assess the internal validity and reliability of the survey instrument and (2) evaluate item fit statistics, including infit and outfit measures, to determine whether individual survey items align with the Rasch model assumptions. The validation of the newly developed College Student Food and Nutrition Security Module (CS-FNSSM) will ensure that it accurately measures the intended construct, enabling the estimation of food and nutrition security status in college students.

## 2. Materials and Methods

### 2.1. Study Design

A mixed-methods research design was employed to validate the CS-FNSSM. This research design combined online surveys with cognitive interview insights from study participants. The design provided a comprehensive evaluation of the psychometric validity and reliability of the CS-FNSSM.

### 2.2. Setting

Participants for this study were recruited using a convenience sampling method of students in selected 4-year universities across the United States. Recruitment efforts included sending email invitations directly to students via their institutional email addresses and partnering with course instructors who distributed the study information to their classes. This approach facilitated access to a broad range of students across different academic programs. This study was conducted among students aged 18 and above, recruited from thirteen public universities across eleven states in the U.S. Study participants from Auburn University, Eastern Kentucky University, University of Rhode Island, University of North Carolina at Charlotte, University of Texas at El Paso, Texas A&M University, University of Maine, Rutgers University, Louisiana State University, University of Tennessee Knoxville, Kansas State University, University of Missouri Columbia, and The Ohio State University participated in the validation of the CS-FNSSM. The diverse student population ensures a wide range of perspectives and experiences, enriching the data and enhancing the validity of the findings. Additionally, the sample promotes equity and inclusion, ensuring that the survey accurately reflects the varied realities of college students across different geographic and demographic contexts, making the results more generalizable and applicable to a broader population.

### 2.3. Quantitative Study Sample

During the Spring and Fall of 2024, an online Qualtrics survey was distributed to participating institutions. The survey included socio-demographic questions (e.g., age, student status, gender, ethnicity), the 2-item USDA food sufficiency screener, and the 13-item College Student Food and Nutrition Security Survey Module (CS-FNSSM). At the end of the survey, participants were asked if they were willing to take part in a test–retest reliability study. Those who agreed were redirected to a separate link, where they provided the last five digits of their phone number. This method maintained participant anonymity while enabling researchers to match responses between the initial survey and the retest. Participants who consented received a follow-up survey link via email seven days after completing the initial survey. Up to three reminder emails were sent within 14 days to encourage completion. Participants who completed both the initial and follow-up surveys were entered into a raffle to win one of ten $25 electronic gift cards. Participants recruited via course instructors also earned extra credit for survey completion.

### 2.4. Instrument

The quantitative survey included demographic questions, a 2-item food sufficiency screener, and the newly developed 13-item College Student Food and Nutrition Security Survey. There were six demographic questions, including age, gender, race, student status, first-generation student status, sexuality, and student employment status. The 2-item food sufficiency is a validated instrument as a quick screener for identifying households at risk of food insecurity. It consists of two key questions derived from the U.S. Department of Agriculture’s 18-item Household Food Security Survey Module. The questions are (1) “Within the past 12 months, we/I worried whether our/my food would run out before we/I got money to buy more” and (2) “Within the past 12 months, the food we/I bought just didn’t last, and we/I didn’t have money to get more.” An affirmative response of “often true” or “sometimes true” to either question is considered a food insecurity risk, while answering “never true” to both questions suggests food security. The measure has been validated in clinical and community settings. It is known for its sensitivity and ease of use, which makes it ideal for settings such as healthcare, schools, and social service programs [25]. The two-item measure was used as a reference tool in validating the CS-FNSSM.

The CS-FNSSM consists of thirteen items, four of which were used to measure food security and four for nutrition security. Respondents were asked to reflect on their experiences during the current semester. Food and nutrition security was scored according to the USDA Guide for Measuring Household Food Security, low scores on the scale indicated little to no food and/or nutrition challenge, classifying the individual as food and nutrition secure. Conversely, higher scores reflect greater levels of food and/or nutrition insecurity. A detailed breakdown of the CS-FNSSM questions and corresponding responses is provided in Table 1.

### 2.5. Qualitative Study Sample

An independent sample of college students was recruited in the Spring of 2025 to participate in a virtual cognitive interview as part of the validation process. Participants first completed the online Qualtrics survey, which included demographic questions and the newly developed survey. To maintain anonymity, they were asked to choose a pseudonym, which was later used to identify them during the virtual interview. Upon completion, they were redirected to the Calendly platform to schedule cognitive interviews. The semi-structured cognitive interviews were conducted via Zoom platform (version: 6.3.11) by a trained researcher and moderators in a private location. The researcher described the study goals, obtained verbal consent, and received permission to record the session. Survey questions were shared on the screen for participants. Participants were recruited through flyers, random selection of students from the university directory, and emails sent to course instructors, who then shared the study information with their classes. Participants who completed the initial survey and the cognitive interview received a $25 electronic gift card. Additionally, participants who were recruited by their course instructors received extra credit for completing the survey.

### 2.6. Internal Validity (Rasch Model)

The Rasch model, a single-parameter logistic item response theory, is frequently used in validation studies to evaluate the internal consistency and measurement properties of survey instruments [26,27]. The Rasch model aids in examining the psychometric properties of survey items and their capacity to measure the same underlying trait, particularly the severity of food and nutrition insecurity. Applying the Rasch model to the CS-FNSSM helps to determine whether the instrument effectively differentiates between varying levels of food security and/or nutrition security while maintaining statistical reliability and validity.

The Rasch model generated estimates of severity parameters for each item and response, fit statistics for each item, and residual correlations between items [26]. The model defines a precise mathematical framework for ranking item severity and offers statistical techniques to estimate the relative impact of an item. It also evaluates whether observed response patterns align with model assumptions. The Rasch model approach is often adopted to assess the effectiveness of food security items and scales [16,24,27,28,29].

The CS-FNSSM items were dichotomized to indicate the presence or absence of food and/or nutrition security and fitted to the Rasch model using conditional maximum likelihood methods. The analysis considered the sensitivity and specificity of survey items in assessing the food and nutrition security experiences. Item-infit and item-outfit statistics were utilized to evaluate how each survey item related to the respondents’ experiences of food and nutrition insecurity. The Rasch model measures the extent to which observed responses align with expectations based on the model’s assumptions. A perfect fit yields a value of 1 for both fit statistics. Scores higher than 1 mean there were more surprising answers than expected. For example, if someone who is food secure answers in the affirmative to a serious question (such as going a whole day without eating), or a food insecure individual saying ‘no’ to a basic question (like worrying about running out of food). Conversely, values below 1 suggest fewer unexpected responses than anticipated, indicating a closer alignment between observed and expected patterns.

The infit statistic helps check how well a survey question fits the expected pattern by giving more attention to answers from people whose level of food insecurity is close to the difficulty of the question. It is especially good at spotting whether people are answering consistently when the question is well-matched to their situation. In contrast, the outfit statistic assesses all answers equally and is more sensitive to unusual or unexpected responses. For example, when someone with very low or very high food insecurity gives an answer that does not match the question’s severity. The Rasch model assumes that all items measure the same underlying concept. Therefore, when an infit value deviates significantly from 1.0, it may suggest that the item does not align well with the construct being measured and may not be suitable for inclusion in the scale. Optimal infit values lie between 0.8 and 1.2, but values from 0.7 to 1.3 are typically still acceptable. Values above 1.0 suggest the answers are more unpredictable than expected, possibly due to random noise or confusion. The same principles apply to the outfit value. However, the outfit value is more affected by a few extreme or surprising answers [27,30].

### 2.7. Reliability Testing

The test–retest method was used in this study to administer the same measure to the same respondents under the same conditions on two separate occasions and the scores were correlated. The survey link was sent on the seventh day following the initial completion date to participants who had consented to take part in the test–retest study. This time interval was selected as considered suitable for participants; a longer interval could result in changes in their circumstances, while a shorter one might cause respondents to remember their initial responses [31,32]. To determine internal reliability, Pearson’s correlation between scores of the two sittings will be measured (r > 0.80 is a strong correlation) [31,32,33].

### 2.8. Statistical Analysis

#### 2.8.1. Data Preparation

The data collected was exported to Microsoft Excel (version 2502) for initial cleaning. This process involved removing incomplete entries, duplicate responses, and data from participants who did not meet the study eligibility criteria.

#### 2.8.2. Analytical Approach

All statistical analyses were conducted using R studio (version 4.4.1) for Windows. Statistical significance was determined at *p* ≤ 0.05. A logistic regression model was employed to evaluate the relationship between food and nutrition security and sociodemographic factors. Food and/or nutrition security was classified as 0–2 affirmative responses, while food and/or insecurity was classified as 3 or more affirmative responses. For continuous scoring, food and/or nutrition security was defined as high (0 affirmative responses), marginal (1–2 affirmative responses), low (3–5 affirmative responses), and very low (6 or more affirmative responses). Food security was treated as a categorical independent variable and analyzed for various demographic characteristics using logistic regression, and for different food groups using linear regression. All models included a dummy variable representing the survey to account for differences in food security measurement and other unobserved variations between populations. Similar classifications were used for both food and nutrition security.

Cognitive interview recordings were transcribed using Zoom transcription. The researcher cross-checked the transcripts for accuracy. A thematic coding process was employed to develop themes and codes, relying on a systematic, researcher-driven approach to data analysis. Two trained research team members (R.F and R.G) independently coded the first four transcripts. Interrater reliability (Cohen’s Kappa of 0.61–0.80 was considered acceptable) and agreement between coders (≥80%) were calculated [34]. After establishing reliability and agreement, coders met to discuss and plan the coding of all interview transcripts using the established codebook. Subsequently, periodic meetings were held to address disagreements and any new codes that arose.

### 2.9. Ethical Considerations

The Institutional Review Board (IRB) of Auburn University approved the study protocol. Participants reviewed the survey information letter attached to the invitation email, which served as virtual consent for participating in the online survey. Prior to each virtual cognitive interview, participants provided verbal consent to confirm their willingness to participate and to allow the session to be recorded. Steps were taken to ensure anonymity and confidentiality, and no identifying information was collected that could link responses to individual participants.

## 3. Results

### 3.1. Characteristics of Respondents

A total of 1234 college students from thirteen 4-year institutions responded to the survey. Due to incomplete responses, 281 submissions were excluded from the analysis. The data presented in this analysis include responses from 953 college students. The majority of participants identified as female (77%) and Caucasian (77%). The majority of the respondents were classified as undergraduate students (97.2%), and 78.9% (*n* = 752) were not first-generation college students. Over half of the participants reported being employed: 48.9%; *n* = 466) reported being employed part-time; 5% (*n* = 49) reported being employed full-time. Table 2 provides a detailed breakdown of the study’s participants.

### 3.2. Rasch Analyses Item Fit

Item-level fit statistics were analyzed to evaluate the alignment of individual items with the Rasch measurement model, using Outfit and Infit Mean Square (MnSq) values. For the Food Security items, Food description (CSFS1) demonstrated an acceptable fit, with an Outfit of 0.452 and an Infit of 0.911. Food quantity (CSFS4) also showed good fit (Outfit = 0.604, Infit = 0.836), while food economics (CSFS5) yielded lower values (Outfit = 0.247, Infit = 0.596), indicating highly consistent responses. The strongest fit was observed for Meal plan (CSFS3), with an Outfit of 0.771 and an Infit of 0.999, both closely aligning with the model’s expectations.

For the Nutrition Security items, nutrition access (NS1) and nutrition quality (NS3) exhibited a good fit, with Outfit values of 0.795 and 0.736 and Infit values of 0.902 and 0.825, respectively. Nutrition health (NS4) also aligned well with the model (Outfit = 0.843, Infit = 0.997). Nutrition habit (NS5) showed lower fit statistics (Outfit = 0.388, Infit = 0.628). See Table 3 for Rasch infit and outfit statistics.

The combined CSFNSSM model (both food and nutrition security) with a threshold of 1.7, prioritizing sensitivity over specificity, the classification results were compared to the reference standard (2-item screener). The model correctly identified 49 insecure cases but misclassified 6 secure cases as insecure (false positives). It failed to detect 200 insecure cases, labeling them as secure (false negatives), while accurately identifying 642 secure cases. This resulted in a sensitivity of 89.09%, demonstrating a strong ability to detect insecure cases, which aligns with the model’s goal to maximize sensitivity. The specificity was 76.25%, indicating a moderate rate of correctly identifying secure cases. The overall classification error rate was 22.97%. With a threshold of 0.6 for Food Security (FS) model alone, the classification model’s performance was evaluated against the 2-item screener. The model correctly identified 72 insecure cases but misclassified 15 secure cases as insecure (false positives). It failed to detect 175 insecure cases, labeling them as secure (false negatives), while accurately identifying 628 secure cases. This resulted in a sensitivity of 82.76%, demonstrating a strong ability to detect insecure cases, and a specificity of 78.21%, indicating a moderate rate of correctly identifying secure cases. The overall classification error rate was 21.35%. When compared to the 2-item food security screener, the Nutrition Security (NS) model demonstrated a sensitivity of 78.18%, correctly detecting 78.18% of participants who were indeed nutrition insecure, with a threshold of 2.97. 74.49% of individuals who were nutrition secure were appropriately identified by its 74.49% specificity. The percentage of people that were misclassified was shown by the total classification error rate, which was 25.29%. The approach accurately identified 654 people who were actually nutrition secure and 43 people who were truly nutrition insecure. Nevertheless, it also incorrectly identified 12 secure people as insecure and 224 nutrition-insecure people as secure. See Table 4 for Rasch model performance against the reference standard, the 2-item screener.

Based on the established Rasch model, thresholds (raw score of survey response) for FNS, FS, and NS were specified to determine prevalence that best fit the model. For FNS, with a threshold total of 5, the Rasch model identified 165 insecure cases and 732 secure cases, with no misclassifications (0 false positives and 0 false negatives). Similarly, for FS, with a threshold total of 2, the Rasch model identified 87 insecure cases and 810 secure cases, again with no misclassifications (0 false positives and 0 false negatives). These results indicate that the Rasch model perfectly classified all cases for both FNS and FS at their respective thresholds. The Rasch model classification for the Nutrition Security (NS) with threshold total of 2, the model correctly identified 55 true insecure cases but misclassified a substantial number of secure individuals, labeling 310 secure cases as insecure (false positives). Notably, the model produced no false negatives, accurately identifying all 568 true secure cases as secure. This resulted in a 100% sensitivity, indicating perfect detection of insecure cases. However, this high sensitivity comes at the cost of reduced specificity due to the significant number of false positives.

Descriptive analyses of food and nutrition security (FNS) levels indicated that most participants were classified as food secure. Figure 1 and Figure 2 illustrate food and nutrition prevalence, and food security and nutrition security prevalence, respectively.

The study investigated the relationship between food security (FS) levels and racial identity using Pearson’s Chi-squared test. Results showed a significant association between race and FS levels (*p* = 0.034), with the highest number of individuals classified as “secure” among Caucasian/White participants. A logistic regression model showed that the Caucasian/White group had a significantly lower likelihood of food insecurity (*p* < 0.001) compared to other racial groups. A marginally non-significant result was seen between FS levels and student status (*p* = 0.071), with only seniors showing a statistically significant coefficient (*p* = 0.023). First-generation students were more likely to experience lower levels of food security compared to their non-first-generation peers (*p* < 0.001). Another significant association was found between gender and FS levels (*p* = 0.019), with females having higher odds of food insecurity compared to males. Neither male nor non-binary gender identities showed statistically significant associations in this model, highlighting gender disparities in food security.

A significant relationship between employment status and food and nutrition security levels (*p* = 0.012) was determined. Part-time workers and those without jobs had the highest respondents in secure and marginal food security categories. However, part-time workers reported more instances of very low food security. Full-time employment showed a potential trend toward greater food security, but this did not reach conventional significance levels (*p* = 0.0742). Overall, employment, especially full-time work, may offer some protection against food insecurity, while unemployment increases vulnerability.

The analysis of the remaining CSFNSSM questions, specifically CSFS2, CSFS3a, and NS2, aimed to determine their association with food and nutrition. Table 1 provides a detailed description of each question. Food and nutrition security levels plan by student meal plan status were evaluated, which is illustrated in Figure 3. The study also looked at the relationship between meal plan status and food and nutrition security (FNS). The results indicated no statistically significant correlation between food and nutrition security (FNS) and the type of meal plan (*p* = 0.081).

Participants reported several reasons that made it difficult for them to access nutritious food. The top three reasons were: healthy foods being too expensive; limited time to shop for groceries; and not knowing how to cook. See Figure 4 for the illustration.

Reliability Testing: Of the One hundred and fifty-nine (*n* = 159) who completed the retest, only 69 students could be linked to their original surveys, and their results were used for the reliability test. The test–retest reliability results indicate that the correlation of scale scores over two time points was 0.59, representing approximately 60% of the responses that are consistent between the first and second administrations of the test. For internal consistency, Cronbach’s alpha was calculated separately for each time point. At Time 1, alpha was 0.82, reflecting good reliability, while at Time 2, it was 0.72, indicating acceptable reliability. Generalizability coefficients were also examined. The between-person reliability was 0.53, while within-person reliability was high at 0.84 (SD = 0.09). Additionally, the internal reliability of the initial survey responses was assessed, resulting in a Cronbach alpha of 0.793, an indication of good internal consistency. The data set consists of 8 items and 897 sample units. With a 95% confidence interval (CI) obtained through bootstrapping with 1000 samples, of 0.770 to 0.814.

### 3.3. Cognitive Interviews

During the study period (January to April 2025), 122 college students responded and completed the online Qualtrics survey, which consisted of 9 demographic questions and the CS-FNSSM question. On average, participants were 22.61 ± 7.06 years old (range 18–53 years). Most participants were female (76%), and slightly more than half were Caucasian (54.4%). Of the participants, 79.5% were undergraduates, while 20.5% were graduate students. Of the 122 students, 30 (18 female and 12 male) participated in the semi-structured cognitive interview. Table 2 provides a further breakdown of participants’ demographics. Coders demonstrated an overall percent agreement of 75%, with a Cohen’s Kappa of 0.65, indicating substantial agreement [33].

The analysis of participants responses from the cognitive interview revealed several interconnected themes that illuminate the complexity of food experiences among college students. After data analysis, we identified the following specific themes: (1) comprehension; (2) demographics/relatability; (3) emotional and psychological; (4) access and barriers; (5) meal plan/institutional influence; (6) food practices; (7) quality and quantity; (8) understanding food insecurity; and (9) nutrition awareness. The themes, sub-themes, and illustrative quotations are listed in Table 5.

## 4. Discussion

This study evaluated the validity and reliability of the newly developed College Student Food and Nutrition Security Survey Model (CS-FNSSM) by comparing it to the established 2-item food sufficiency screener. Importantly, this study is the first to propose and validate a food and nutrition security model specifically tailored to college students in the United States. By psychometrically assessing the CS-FNSSM, this research provides a valuable tool to evaluate food and nutrition security from the perspective of this unique population. Overall, the instrument demonstrated acceptable psychometric properties. The CS-FNSSM evaluates two key constructs: food security, which reflects access to sufficient food, and nutrition security, which captures the quality and adequacy of food necessary for a healthy life [35,36]. The model fit was acceptable, the individual items indicated a good model fit with all items falling within MnSq < 1.5 [37,38]. Among the Food Security (CSFS) items, food description (CSFS1), food quantity (CSFS4), and food economics (CSFS5) demonstrated an acceptable fit, with food economics (CSFS5) showing the lowest MnSq values (Outfit = 0.247, Infit = 0.596), suggesting potential overfit, which may indicate redundancy or highly predictable responses. Meal plan (CSFS3) exhibited the strongest alignment with the model, with an Outfit MnSq of 0.771 and an Infit MnSq of 0.999. Similarly, the Nutrition Security (NS) items showed a good fit, with Nutrition access (NS1), nutrition quality (NS3), and nutrition health (NS4) all falling within the optimal range. Nutrition habit (NS5), while slightly overfitting (Outfit = 0.388, Infit = 0.628), still met the model’s criteria. These findings suggest that the items function well within the Rasch framework, supporting the structural validity of the CS-FNSSM. Also, its ability to reliably measure food and nutrition security among college students.

The Rasch model-based classification demonstrated perfect accuracy for the combined FNS and FS models at their respective thresholds (5 for FNS and 2 for FS), with no false positives or false negatives. Suggesting the Rasch-derived thresholds may offer a more precise method for identifying food and nutrition insecurity. The NS Rasch model, despite achieving 100% sensitivity, incorrectly classified 310 secure individuals as non-secure (false positives), leading to a significant drop in specificity. This trade-off highlights the challenge of balancing sensitivity and specificity in classification models when the goal is to minimize the risk of overlooking insecure individuals.

The descriptive analyses of the combined food and nutrition status revealed that over two-thirds of respondents, approximately 82%, fell into the secure category, and 18% were classified as insecure. In contrast, Nutrition Security (NS) indicated a more concerning picture: only 61% of participants were classified as secure, while 39% were considered nutrition insecure. This relatively high level of nutrition insecurity suggests that many students may have access to enough food, but not necessarily nutritionally adequate or health-promoting. Interestingly, the Food Security (FS) analysis showed that 10.7 percent and 89.3 percent of participants were food insecure and secure, respectively. The discrepancy between FS and NS outcomes highlights a critical distinction: while most students may not experience hunger or food scarcity, a significant portion may still lack access to nutritious, balanced meals. These findings emphasize the importance of assessing food and nutrition security as distinct but related constructs. Relying solely on traditional food security measures may underestimate the challenges students face in achieving overall dietary well-being [9,19,22,39]. As such, using a psychometrically sound tool like the CS-FNSSM is vital for effectively assessing college students’ experiences with food and nutrition security.

The test–retest reliability coefficient of 0.59 observed in this study suggests a moderate consistency in the instrument’s ability to measure food and nutrition security over time [40,41]. While this value does not meet the conventional threshold of 0.70 often cited for “acceptable” reliability, it remains within an acceptable range, considering the variability of food and nutrition security, due to seasonal, economic, or individual-level factors [31,42]. Moreover, the length of time between test administrations can influence reliability estimates. In this context, a coefficient of 0.59 may reflect both the real-world variability in food and nutrition security and the instrument’s sensitivity to such changes [31]. Additionally, the internal consistency of the scale as measured by Cronbach alpha (*α* = 0.793), was within the acceptable range [43]. A bootstrap 95% confidence interval (0.770–0.814) further confirmed the stability and robustness of this estimate across different samples [44].

Cognitive interviews provided valuable insights into how college students interpreted and related to the items in the CS-FNSSM. Participants generally demonstrated strong comprehension, recognizing key terms related to food insecurity and connecting them to their personal experiences. For example, phrases like “cut the size” or “skip meals” due to financial constraints were well understood and often paraphrased in relatable terms, such as adjusting portion sizes based on meal plan limitations. The demographic relevance of the items was also affirmed, with students noting that the questions reflected common challenges faced by peers, particularly those without access to kitchens or reliable transportation. Emotional responses varied: while some students reported no stress around food access, others expressed long-term concerns about the health impacts of their dietary habits.

Barriers such as limited transportation and time constraints emerged as significant factors affecting students’ ability to obtain and prepare nutritious meals. Many participants described a heavy dependence on institutional meal plans, which, while helpful, often lacked flexibility or coverage for healthier food options like fresh produce. Students also shared food prioritization strategies, frequently relying on convenience foods due to time pressures or limited nutritional knowledge. While some reported access to a variety of foods, others described a mismatch between available and preferred options, often settling for what was accessible rather than ideal. College students’ understanding of food insecurity varied, with some recognizing it only in hindsight or the experiences of others. Finally, the theme of nutrition awareness revealed that while many students aimed to make healthy choices, their definitions of “healthy” or “unsafe” food were shaped by personal beliefs rather than formal guidelines. These findings underscore the importance of culturally and contextually relevant survey language and highlight the need for tools like the CS-FNSSM to capture the nuanced realities of food and nutrition security among college students.

While this study utilized the newly developed CS-FNSSSM, it is essential to consider its relationship with the established USDA FSSM. Both instruments aim to assess aspects of food access, but they differ in scope and target population. The USDA FSSM primarily focuses on household-level food security, emphasizing economic access to sufficient food. In contrast, the college student-focused survey incorporates additional dimensions relevant to college populations, such as meal plan, reference period food and nutrition experience “ this semester”, and barriers to nutritious food access. These distinctions tailor the tool to the unique experiences of students, while maintaining conceptual alignment with national standards. Recognizing these similarities and differences is vital for interpreting the findings and situating the new instrument within broader food security research frameworks.

Students face unique challenges that significantly impact their academic performance and well-being, including financial constraints, limited access to adequate and nutritious food, competing academic demands, and the ongoing process of learning how to navigate adulthood. Existing literature has highlighted the prevalence of food insecurity and poor dietary patterns among students [17,18,19,21,22]. Many studies rely on generalized tools that may not fully capture the lived experiences or may be misinterpreted by students [9,19,21,22]. This research contributes to the field by presenting a survey instrument specifically tailored to the student population, grounded in psychometric validation to ensure reliability and relevance. By focusing on both the content and structure of the survey, the study addresses gaps in previous research. [18,19,45] Provides a measure that evaluates nutritional quality, not just food availability [45], that can inform campus health initiatives, policy interventions, and future research in public health nutrition.

### Strengths and Limitations

This validation study employed a multi-method approach to evaluate a newly developed survey instrument, incorporating quantitative and qualitative strategies. The survey was distributed to a large sample across institutions in the U.S. The wide institutional reach enhanced the generalizability of the findings by capturing diverse student populations and educational contexts. The Rasch model provided strong evidence for construct validity, offering detailed insights into item functioning and scale structure. Reliability was assessed through internal consistency measures, while cognitive interviews with college students added depth by evaluating how students interpreted and responded to the items, supporting response process validity.

However, the study also had limitations. Despite the large sample recruited, the predominantly White and female study participants limit the generalizability of the findings. Additionally, potential sampling bias and uneven response rates across institutions may have affected how well the sample reflected the broader population. The Rasch model, while robust, assumes unidimensionality, which may not fully capture the complexity of certain constructs measured by the survey. Also, the limited number of cognitive interviews, although data saturation was established, may not have been sufficient to identify all issues related to item clarity and interpretation across diverse student groups. Although data was not collected on students’ familial economic status, student employment status was collected, which is equally relevant in determining student food and nutrition needs.

## 5. Conclusions

The CS-FNSSM is a validated instrument designed to assess food and nutrition security among college students. It demonstrates strong reliability and validity, making it a promising tool for research aimed at estimating college students’ food and nutrition status and informing targeted interventions. Key strengths of the instrument include its use of a large, diverse sample across multiple universities, a robust study design, the Rasch model, high sensitivity, and strong psychometric properties. Notably, this is the first validated tool to measure both food and nutrition security specifically in college populations, addressing a critical gap in existing research. Continued validation for nutrition security items using a validated nutrition security screener will further support the CS-FNSSM. Future prospective studies are recommended to evaluate the instrument’s predictive capability in tracking changes in students’ food and nutrition security over time.

## Figures and Tables

**Figure 1 nutrients-17-02514-f001:**
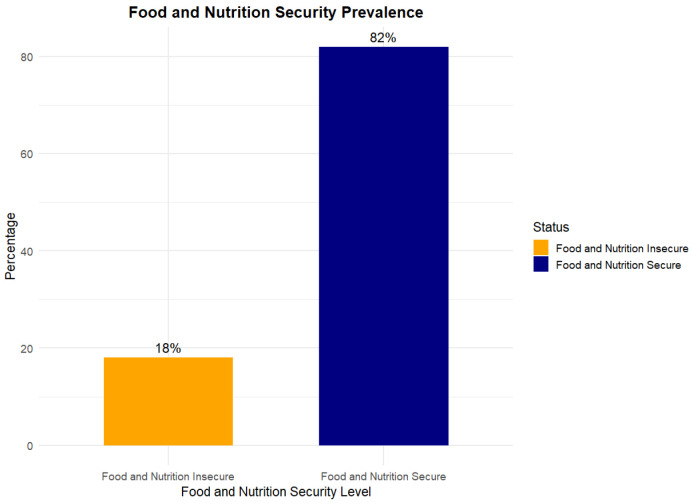
Food and nutrition security prevalence.

**Figure 2 nutrients-17-02514-f002:**
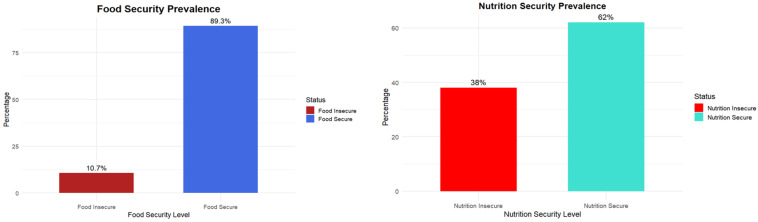
Food security prevalence (**right**) and nutrition security prevalence (**left**).

**Figure 3 nutrients-17-02514-f003:**
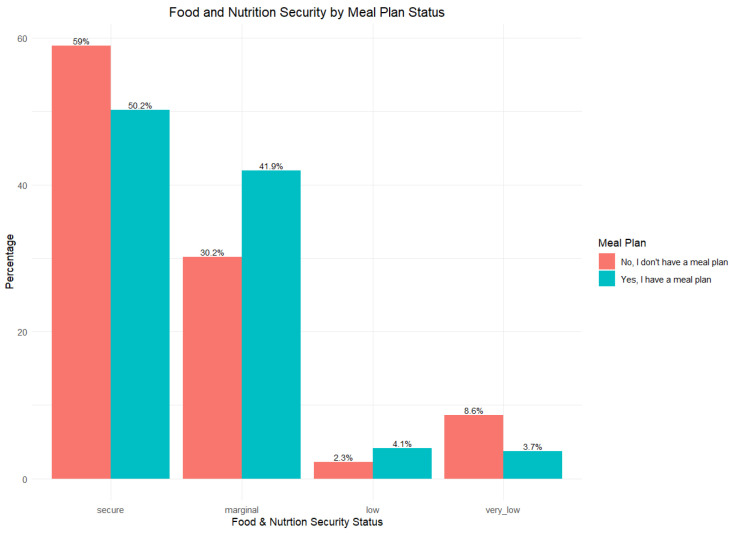
Food security levels by meal plan status: Students with a meal plan were significantly more likely to be food secure than those without one (*p* < 0.0001). The data show that while both groups had similar numbers in the “secure” category, students without a meal plan were more likely to fall into the “very low” food security category. The chi-square test confirms that this difference is unlikely due to chance.

**Figure 4 nutrients-17-02514-f004:**
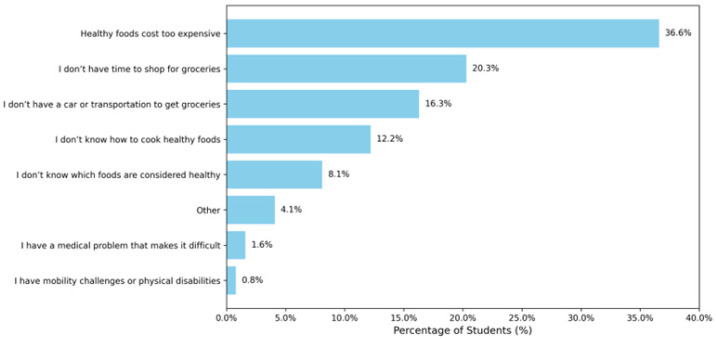
Challenges in accessing healthy foods.

**Table 1 nutrients-17-02514-t001:** College student food and nutrition security survey module questions.

Question	Response Options
CS-FS1 (Food description): Which of these statements best describes the food that you have eaten this semester?	Enough of the kinds of food I want to eat. Enough but not always the kinds of food I want. * Sometimes not enough to eat. * Often not enough to eat. Refused to answer.
CS-FS3 (Meal Plan): Option 1 (no meal plan): The first statement is “This semester, I worried whether my food would run out before I got money to buy more” OR Option 2 (with meal plan): The first statement is “I worried whether my meal plan would run out before the end of the semester.”	* Often true. * Sometimes true. Never true. Refuse to answer.
CS-FS4 (Food quantity): In this semester, did you ever cut the size of your meals or skip meals because you didn’t have enough money for food or enough in your meal plan?	* Yes. No [Skip CS-FS5]. Don’t know [Skip CS-FS5].
CS-FS4a. [IF YES ABOVE, ASK] How often did this happen.	* Often (every week). * Sometimes (some weeks) Occasionally (couple of times a semester) Choose not to answer.
CS-FS5 (Food economics): In this semester, did you ever not eat for a whole day because there wasn’t enough money for food?	* Yes. No [Skip NS1]. Don’t know [Skip to NS1].
CS-FS5a. [IF YES ABOVE, ASK] How often did this happen?	* Often (every week). * Sometimes (some weeks) Occasionally (couple of times a semester) Choose not to answer.
NS-1 (Nutritious access): Thinking about this semester, how hard was it for you to regularly get nutritious foods that support your health and well-being?	* Extremely difficult * Moderately difficult * Slightly difficult Neither easy nor difficult Slightly easy Moderately easy Don’t know/prefer not to answer
NS-3 (Nutrition quality): In this semester, I had to eat some foods that were unhealthy because I couldn’t get healthy food.	Never Rarely * Sometimes * Often * Always Choose not to answer.
NS-4 (Nutrition health): In this semester, I worried that the food I was able to eat would hurt my health and well-being.	Never Rarely * Sometimes * Often * Always Choose not to answer.
NS-5 (Nutrition habits): In this semester, I had to eat the same unhealthy thing for several days in a row because I didn’t have money to buy other food.	Never Rarely * Sometimes * Often * Always Choose not to answer.
Other Survey Items	
CS-FS2. Are you on a meal plan?	Yes, I have a meal plan [move to CS-FS3] No, I don’t have a meal plan [move to option #2]
CS-FS3a. What level of meal plan do you have?	Comprehensive (most meals covered) Partial (some meals) Other, please explain
NS-2. There are several reasons why people can’t eat healthy foods as much as they think they should. In this semester, which of these limited your ability to eat healthier meals? [Select all options that apply]	Healthy foods are too expensive. I don’t have a car or transportation to reach stores or food pantries that have healthy foods. I don’t know which foods are considered healthy foods. I don’t know how to cook healthy foods. I don’t have time to shop for groceries. I have mobility challenges or physical limitations that make it difficult for me to prepare healthy foods. I have a medical problem that makes it difficult for me to eat healthy foods. Other (please specify)

Note. * denotes an affirmative response. CS-FS: College student food security, NS: nutrition security.

**Table 2 nutrients-17-02514-t002:** Participant characteristics.

	Quantitative Survey (*n* = 953)	Survey Prior to Qualitative Study (*n* = 122)
Age (mean (SD))	19.85 (2.71)	22.61 (7.06)
Gender (%)		
Female	737 (77.3)	93 (76.2)
Male	206 (21.6	29 (23.8)
Non-binary	10 (1.0)	
Race (%)		
Asian	67 (7.0)	21 (17.4)
Black/African American	77 (8.1)	25 (20.7)
Caucasian/White	736 (77.2)	66 (54.5)
Hispanic/Latino	50 (5.2)	2 (1.7)
Native American	1 (0.1)	2 (1.7)
Prefer not to say	12 (1.3)	4 (3.3)
Two or more, please specify	10 (1.0)	1 (0.8)
Student status (%)		
Graduate student	26 (2.7)	25 (20.5)
Undergraduate	927 (97.3)	97 (79.5)
Freshman (1st year undergraduate)	238 (25.0)	
Sophomore (2nd year undergraduate)	302 (31.7)	
Junior (3rd year undergraduate)	226 (23.7)	
Senior (4th year undergraduate)	161 (16.9)	
First-generation student (%)		
Yes, I am	193 (20.3)	38 (31.1)
No, I am not	752 (78.9)	84 (68.9)
I do not Know	8 (0.8)	
Sexuality (%)		
Bisexual	69 (7.2)	8 (6.6)
Heterosexual or straight	862 (90.5)	105 (86.1)
Homosexual	10 (1.1)	7 (5.7)
Other, please specify	11 (1.2)	2 (1.6)
Employment status (%)		
Full-time (36+ h per week)	49 (5.1)	5 (4.1)
Part-time (1–36 h per week)	466 (48.9)	73 (59.8)
I do not have a job	416 (43.7)	39 (32.0)
Prefer not to answer	22 (2.3)	5 (4.1)

**Table 3 nutrients-17-02514-t003:** Rasch infit and outfit statistics for items in the CS-FNSSM, based on survey responses.

	Item	Outfit Item	Infit Item
Food description (CSFS1)	CSFS1	0.4520	0.9106
Food quantity (CSFS4)	CSFS4	0.6044	0.8356
Food economics (CSFS5)	CSFS5	0.2467	0.5959
Nutrition access (NS1)	NS1	0.7948	0.9021
Nutrition quality (NS3)	NS3	0.7355	0.8245
Nutrition health (NS4)	NS4	0.8430	0.9969
Nutrition habit (NS5)	NS5	0.3878	0.6279
Meal plan (CSFS3)	CSFS3	0.7713	0.9994

**Table 4 nutrients-17-02514-t004:** Rasch model performance comparison (vs. 2-item screener).

Model	Threshold	Sensitivity	Specificity	False Positives	False Negatives	Correctly Classified	Error Rate
Combined (CSFNSSM)	1.7	89.09%	76.25%	6	200	691 (49 insecure + 642 secure)	22.97%
Food Security (FS)	0.6	82.76%	78.21%	15	175	700 (72 + 628)	21.35%
Nutrition Security (NS)	2.97	78.18%	74.49%	12	224	697 (43 + 654)	25.29%

**Table 5 nutrients-17-02514-t005:** Themes, sub-themes, and illustrative quotations.

Theme	Sub-Theme	Ilustrative Quotation
Comprehension	Direct Understanding	I found it easy to answer the question, because I am very passionate about health and nutrition.
	Keyword match	Key terms here are nutritious food, well-being, and food that supports your health.
Demographics/Relatability	Demographic Relevance	“this semester” is an important one. I’ve seen other food security screeners that ask about the past year, but I like that this one focuses on the current semester.”
	Relatability to College Students	Absolutely, yes, I think college students can definitely relate. Many students are on meal plans, and some, like me, also get money from parents or work to support themselves. So, it’s definitely beneficial to know this.
Emotional and Psychological	Lack of Food-Related Stress	I have access to the food, and I can always order online.
	Emotional Response	Maybe I have a lot of bills to pay in some months, and I might, just have be on a strict budget…. yes sometimes it’s because I’m worried my foodstuffs will run out before I get money to buy more.
Access and Barriers	Lack of Transportation	I don’t have a car or transportation to reach stores or food pantries.
	Time Constraints	I might choose something unhealthy because I don’t have time to get something else.
Meal Plan/Institutional Influence	Dependence on Meal Plan	I have a meal plan that allows me to have decently healthy foods, unprocessed.
	Level of Meal Plan Impacts Experience	I have an unlimited meal plan, so I can get as much nutrients as I want, but I often like to resort to Burger patties.
Food Practices	Food Prioritization Strategies	It’s hard to know which foods are truly healthy, so having knowledge about that is important.
	Frequent Convenience Meal Consumption	I’ll go for a burger instead, because I’m not able to go to the store to buy, maybe foods and make a smoothie.
Quality and Quantity	Variety of Healthy Foods	Never. Thinking about what I’ve eaten, there was never a time when I had to eat something like ramen for a whole week. I always had a choice between different options.
	Lack of Food Quality	Sometimes… the food being a little undercooked… a little dry or stringy.
Understanding Food Insecurity	Severity of Food Insecurity	I don’t skip meals, but I might at times reduce a portion of my food.
	Experience of Food Insecurity	There are definitely points during the semester, like I mentioned before, where I’m just trying to stretch my groceries.
Nutrition Awareness	Understanding the Definition of Healthy Foods	The only thing I’d point out is that different people may interpret what counts as “healthy” or “unhealthy” differently. But having a definition before the question really helps clarify it.

## Data Availability

Data included in the article are referenced in the article.

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
