# Peer review of "Psychometric Validation of a Standardized Instrument for Assessing Food and Nutrition Security Among College Students"

_nutrients, 2025, doi:10.3390/nu17152514_

Round 1

Reviewer 1 Report

Comments and Suggestions for Authors

The paper entitled ‘Psychometric Validation of a Standardized Instrument for Assessing Food and Nutrition Security Among College Students’ is a quite interesting story and has the potential to attract readers. Just very minor remarks given below could be addressed.  

General: please carefully check the whole manuscript (body text, tables, figures) for editorial shortcomings, e.g. lack of dots, redundant spaces, text to be deleted (lines 221-223). All figures need improvement. Some results can be combined into one figure, some figure numbers are incorrect, etc. Use percentages instead of student numbers, add labels (in %) into the figures.

Introduction/Discussion:

I suggest adding some information about the similarities/differences between the U.S. Department of Agriculture’s Food Security Survey Module and College Student Food and Nutrition Security Surve.

Results,
Table 1:
For me not all information is clear, e.g. CS-FS1 and options 1 and 2 without giving any references?

It would be good to add the explanation for abbreviations in table note.

I would also add the column with the number of question as the Authors refer to the number in further sections. In some cases, Authors use the expression “question 4” while in other – “CSFS2, CSFS3”. It is not easy to follow the text and understand the results.

Table 2: Please add in the table note the explanations for student status names.

I miss the information about the economic status which is highly important for food security. On the other hand, the sexuality is less important.

Please explain the issue of “the reference standard (2-item screener)” – in Materials and Methods?

Maybe it would be beneficial to add a new table that summarizes the results of Rasch model and correctly/incorrectly identified cases with numbers and percentages?

Figures are too simple or unclear. I cannot see the sense of Fig. 1, 2, etc. And all the comments given before.

The text in lines 325-327 looks improper in this place.  

Please correct: “Figure 4. Meal Plan Status Type; for students who had meal plan most had the comprehensive meal 340 plan where most meals covered, approximately 180 had a partial meal plan and about 30 had other 341 types of meal plan like the athlete meal plan, sorority associated plans.”

Lines 348-363: G3 and G6 should be explained in Statistical analysis section.

Lines 384-441: the text needs some introduction/explanation, etc.

Discussion:

Lines 444-445: “by comparing it to the established 2-item food sufficiency screener” – if it is clearly explained in Methods it would be easier to follow the whole idea/results, etc.

Author Response

Comment and Suggestions:

Response

Reviewer 1:

General: please carefully check the whole manuscript (body text, tables, figures) for editorial shortcomings, e.g. lack of dots, redundant spaces, text to be deleted (lines 221-223). All figures need improvement. Some results can be combined into one figure, some figure numbers are incorrect, etc. Use percentages instead of student numbers, add labels (in %) into the figures.

Completed

Introduction/Discussion:

I suggest adding some information about the similarities/differences between the U.S. Department of Agriculture’s Food Security Survey Module and College Student Food and Nutrition Security Survey

A paragraph was added before Strengths and limitations  in the discussion section. It discusses the similarities/differences of the CS-FSSM and the USDA FSSM.

Table 1:

For me not all information is clear, e.g. CS-FS1 and options 1 and 2 without giving any references?

It would be good to add the explanation for abbreviations in table note.

Explanation added to clarify when to use option 1 and option 2.

Completed.

I would also add the column with the number of question as the Authors refer to the number in further sections. In some cases, Authors use the expression “question 4” while in other – “CSFS2, CSFS3”. It is not easy to follow the text and understand the results.

Updated to refer to the CSFS or NS item being described.

Table 2: Please add in the table note the explanations for student status names.

Explanations added to student status names.

I miss the information about the economic status which is highly important for food security. On the other hand, the sexuality is less important.

Statement added: Although data wasn’t collected on students’ familial economic status, student employment status was collected, which is equally relevant in determining student food and nutrition needs”.

Please explain the issue of “the reference standard (2-item screener)” – in Materials and Methods?

Completed

Maybe it would be beneficial to add a new table that summarizes the results of Rasch model and correctly/incorrectly identified cases with numbers and percentages?

Completed

Figures are too simple or unclear. I cannot see the sense of Fig. 1, 2, etc. And all the comments given before.

The figures were improved, and comments were removed.

The text in lines 325-327 looks improper in this place. 

The statement was removed, and the figure was revised to enhance clarity.

Please correct: “Figure 4. Meal Plan Status Type: for students who had meal plan most had the comprehensive meal 340 plan where most meals covered, approximately 180 had a partial meal plan and about 30 had other 341 types of meal plan like the athlete meal plan, sorority associated plans.”

Figure was taken out , and replaced with Figure 3 which describe Food Security levels by Meal Plan Status

Lines 348-363: G3 and G6 should be explained in Statistical analysis section.

The Reliability Testing section under Results has been rewritten, and G3 and G6 have been removed to enhance clarity.

Lines 384-441: the text needs some introduction/explanation, etc.

The information was omitted to avoid redundancy, as it is already clearly presented in Table 4.

Discussion:

Lines 444-445: “by comparing it to the established 2-item food sufficiency screener” – if it is clearly explained in Methods it would be easier to follow the whole idea/results, etc.

Completed.

Reviewer 2 Report

Comments and Suggestions for Authors

Dear Authors,

I carrefully read the manuscript „Psychometric Validation of a Standardized Instrument for Assessing Food and Nutrition Security Among College Students”. I want to congratulate you for your very successful work. The research aim was to validate a food and nutrition survey among students from thirteen US universities. The research protocol is rigorous established in order to validate the newly developed College Student Food and Nutrition Security Survey.

To enhance the clarity of presented data you should pay attention to these aspects:

  • Abstract: you should eliminate some extra-spaces
  • Line 37: I would say it's inappropriate to declare that food security treat disease, I think it’s sufficient the statement „promote well-being, prevent disease, and, if needed, emphasizing equitable access to healthy...”. So, I suggest to eliminate „treat disease” from this paragraph.
  • Materials and Methods: You should explain also how you ensured that the recruitment of participants was from the thirteen public universities? Have you use a specific plaform within the university (intranet) or the participants institutional e-mails?
  • At the beginning of the manuscript you formulated 4 objectives of your study. Please clarify in the Results section which are the resulted data related to objectives 3 and 4: “(3) examine the association between food and nutrition security status and sociodemographic factors” and „(4) compare the performance of the new survey with standard food security assessment tool.”
  • In the above context, discussion section should be enriched with findings related to objectives 3 and 4. You should highlight in the Discussion section the relevance of your study. Discuss what your research add to the subject area compared with other published materials.
  • Please structure the conclusions so that the main results are presented. Do they address the main questions/ objectives posed at the beginning of the manuscript?
  • Throughout the text: check all the extra-spaces and remove them.

Author Response

Comment and Suggestions:

Response

Reviewer 2

Abstract: you should eliminate some extra-spaces

Completed

Line 37: I would say it's inappropriate to declare that food security treat disease, I think it’s sufficient the statement „promote well-being, prevent disease, and, if needed, emphasizing equitable access to healthy...”. So, I suggest to eliminate „treat disease” from this paragraph.

Completed

Materials and Methods: You should explain also how you ensured that the recruitment of participants was from the thirteen public universities? Have you use a specific platform within the university (intranet) or the participants institutional e-mails?

Clarification has been added regarding the recruitment and sampling process.

At the beginning of the manuscript, you formulated 4 objectives of your study. Please clarify in the Results section which are the resulted data related to objectives 3 and 4: “(3) examine the association between food and nutrition security status and sociodemographic factors” and „(4) compare the performance of the new survey with standard food security assessment tool.”

The study objective was reformulated to focus on the psychometric evaluation of the survey. The other objectives previously stated will be addressed in a subsequent paper.

In the above context, discussion section should be enriched with findings related to objectives 3 and 4. You should highlight in the Discussion section the relevance of your study. Discuss what your research add to the subject area compared with other published materials.

Completed

Please structure the conclusions so that the main results are presented. Do they address the main questions/ objectives posed at the beginning of the manuscript?

Throughout the text: check all the extra-spaces and remove them.

Completed.

Round 2

Reviewer 2 Report

Comments and Suggestions for Authors

Thank you for your concise and clear responses. The manuscript was reviewed according to my suggestions. Your paper is now more accurate and could be considered for publication in Nutrients.